# LinGCN: Structural Linearized Graph Convolutional Network for Homomorphically Encrypted Inference

[*]**Hongwu Peng**[1], [*]**Ran Ran**[2], **Yukui Luo**[3], **Jiahui Zhao**[1], **Shaoyi Huang**[1], **Kiran Thorat**[1],
**Tong Geng**[4], **Chenghong Wang**[5], **Xiaolin Xu**[6], **Wujie Wen**[2], **Caiwen Ding**[1]
[*]These authors contributed equally.
[1]University of Connecticut, [2]North Carolina State University,
[3] University of Massachusetts Dartmouth, [4]University of Rochester,
[5]Indiana University Bloomington, [6]Northeastern University
{hongwu.peng, jiahui.zhao, shaoyi.huang, kiran_gautam.thorat}@uconn.edu,
{rran, wwen2}@ncsu.edu, yluo2@umassd.edu, tgeng@ur.rochester.edu
cw166@iu.edu, x.xu@northeastern.edu, caiwen.ding@uconn.edu

## Abstract

The growth of Graph Convolution Network (GCN) model sizes has revolutionized numerous applications, surpassing human performance in areas such as personal healthcare and financial systems. The deployment of GCNs in the cloud raises privacy concerns due to potential adversarial attacks on client data. To address security concerns, Privacy-Preserving Machine Learning (PPML) using Homomorphic Encryption (HE) secures sensitive client data. However, it introduces substantial computational overhead in practical applications. To tackle those challenges, we present LinGCN, a framework designed to reduce multiplication depth and optimize the performance of HE based GCN inference. LinGCN is structured around three key elements: (1) A differentiable structural linearization algorithm, complemented by a parameterized discrete indicator function, co-trained with model weights to meet the optimization goal. This strategy promotes fine-grained node-level non-linear location selection, resulting in a model with minimized multiplication depth. (2) A compact node-wise polynomial replacement policy with a second-order trainable activation function, steered towards superior convergence by a two-level distillation approach from an all-ReLU based teacher model. (3) an enhanced HE solution that enables finer-grained operator fusion for node-wise activation functions, further reducing multiplication level consumption in HE-based inference. Our experiments on the NTU-XVIEW skeleton joint dataset reveal that LinGCN excels in latency, accuracy, and scalability for homomorphically encrypted inference, outperforming solutions such as CryptoGCN. Remarkably, LinGCN achieves a 14.2× latency speedup relative to CryptoGCN, while preserving an inference accuracy of ~75% and notably reducing multiplication depth. Additionally, LinGCN proves scalable for larger models, delivering a substantial 85.78% accuracy with 6371s latency, a 10.47% accuracy improvement over CryptoGCN. The codes are shared on Github[1].

## 1 Introduction

Graph learning, a deep learning subset, aids real-time decision-making and impacts diverse applications like computer vision [1], traffic forecasting [2], action recognition [3, 4], recommendation systems [5], and drug discovery [6]. However, the growth of Graph Convolution Network (GCN)

---

[1]https://github.com/harveyp123/LinGCN-Neurips23

37th Conference on Neural Information Processing Systems (NeurIPS 2023).

model sizes [7, 8] introduces challenges in integrating encryption into graph-based machine learning services. Platforms such as Pinterest's PinSAGE [9] and Alibaba's AliGraph [10] operate on extensive user/item data graphs, increasing processing time. Furthermore, deploying GCN-based services in the cloud raises privacy concerns [11, 12] due to potential adversarial attacks on client data, such as gradient inversion attacks [13, 14]. Given the sensitive information in graph embeddings, it's crucial to implement lightweight, privacy-focused strategies for cloud-based GCN services.

Privacy-Preserving Machine Learning (PPML) with Homomorphic Encryption (HE) helps protect sensitive graph embeddings, allowing client data encryption before transmission and enabling direct data processing by the cloud server. However, this security comes with substantial computational overhead from HE operations such as rotations, multiplications, and additions. For instance, using the Cheon-Kim-Kim-Song (CKKS) [15], a Leveled HE (LHE) [16] scheme, the computational overhead escalates significantly (e.g. $\sim 3\times$) with the total number of successive multiplications, which is directly related to the layers and non-linear implementations in deep GCNs.

Three research trends have emerged to lessen computational overhead and inference latency for HE inference. First, *leveraging the graph sparsity*, as in CryptoGCN [12], reduces multiplication level consumption. Second, *model quantization*, seen in TAPAS [17] and DiNN [18], binarizes activation and weight to 1 bit, leading to a 3%-6.2% accuracy loss on smaller datasets and limited scalability. Third, *nonliear operation level reduction*, as in CryptoNet [19] and Lola [20], substitutes low-degree polynomial operators like square for ReLU in CNN-based models. However, these methods struggle with performance due to absent encryption context of graph features, and the latter two are not suitable for GCN-based models.

To bridge the research gap, in this paper, we propose LinGCN, an end-to-end policy-guided framework for non-linear reduction and polynomial replacement. This framework is designed to optimize deep Spatial-Temporal Graph Convolution Network (STGCN)-based models for HE-based inference. We conduct extensive experiments on the NTU-XVIEW [21] skeleton joint [22, 23, 24] dataset and compare our approach with CryptoGCN [12]. The pareto frontier comparison of accuracy-latency between LinGCN and CryptoGCN [12] demonstrates

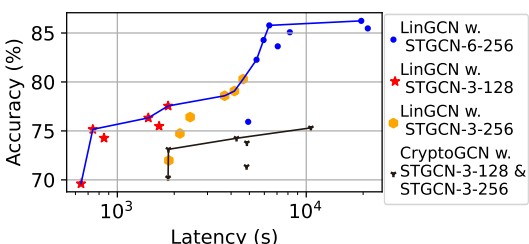

Figure 1: Frontier of LinGCN vs. CryptoGCN [12]

that LinGCN surpasses CryptoGCN [12] by over 5% accuracy under the same latency budget, and achieves a 14.2-fold latency reduction under the same accuracy constraint (~75% accuracy).

Our proposed framework, LinGCN, is principally influenced by two observations: the need to conserve levels in the CKKS scheme and the crucial role of synchronized linearization. In the CKKS scheme, deeper networks demand a larger polynomial degree, thereby increasing the latency of homomorphically encrypted operators. Non-linear layer pruning can mitigate this by reducing level consumption. However, unstructured non-linear pruning is insufficient; structural linearization becomes essential for effective level reduction, underscoring synchronized linearization's significance in optimizing CKKS performance. We summarize our contributions as follows:

1. We present a **differentiable structural linearization** algorithm, paired with a parameterized discrete indicator function, which guides the structural polarization process. This methodology is co-trained with model weights until the optimization objective is attained. Our approach facilitates fine-grained node-level selection of non-linear locations, ultimately yielding a model with reduced multiplication levels.

2. We introduce a node-wise polynomial replacement policy utilizing a second-order trainable polynomial activation function. This process is further steered by an all-ReLU based teacher model employing a two-level distillation approach, thus promoting superior convergence.

3. We have engineered a corresponding HE solution that facilitates finer-grained operator fusion for node-wise activation functions. This development further mitigates multiplication level consumption during our HE-based inference process.

## 2 Background and Related Work

**CKKS Homomorphic Encryption Scheme.** The CKKS scheme [15], based on the ring learning with errors (RLWE) problem, allows arithmetic operations on encrypted fixed-point numbers. It provides configurable precision via encryption noise as natural error $e$ [25]. We denote the cyclotomic polynomial degree as $N$ and the polynomial coefficient modulus as $Q$, both cryptographic parameters. A ciphertext $ct$ encrypts a message $m$ with noise $e$ as $ct = m + e$ (mod $Q$). The CKKS scheme's security level [26], measured in bits, suggests $2^{128}$ operations to break a $\lambda = 128$ encryption. CKKS supports operations including ciphertext addition $Add(ct_1, ct_2)$, ciphertext multiplication $CMult(ct_1, ct_2)$, scalar multiplication $PMult(ct, pt)$, rotation $Rot(ct, k)$ and rescaling $Rescale(ct)$. Scalar multiplication multiplies a ciphertext with plaintext, and rotation cyclically shifts the slot vector. For example, $Rot(ct, k)$ transforms $(v_0, ..., v_{N/2-1})$ into $(v_k, ..., v_{N/2-1}, v_0, ..., v_{k-1})$. The level of a ciphertext $(L)$, the number of successive multiplications it can undergo without bootstrapping, is reduced by the $Rescale$ operation after each multiplication. With the level-reduced target network, we do not consider using bootstrapping [27] in this work.

**STGCN for Graph Dataset with Timing Sequence.** The focus of this study revolves around the STGCN, a highly esteemed category of GCNs proposed by [4]. The STGCN model primarily employs two core operators, Spatial graph convolution (GCNConv) and Temporal convolution, designed to extract spatial and temporal information from input graph data, respectively. The Spatial convolution operation, as defined in Equation 1, precisely illustrates the process of extracting spatial information. In Equation 1, the variables $A$, $D$, $X_i$, $W_i$, and $X_{i,\text{out}}$ correspond to the adjacent matrix, degree matrix, input feature, weight parameter, and output feature, respectively [4].

$$X_{i,out} = D^{-\frac{1}{2}}(A + I)D^{-\frac{1}{2}}X_iW_i \tag{1}$$

**Related Work.** CryptoNets [19] pioneered Privacy-Preserving Machine Learning (PPML) via Homomorphic Encryption (HE) but suffered from extended inference latency for large models. Subsequent studies, like SHE [28] which realizes realize the TFHE-based PPML, and those employing Multi-Party Computation (MPC) solutions[29, 30, 31], reduced latency but faced high communication overhead and prediction latency. Research like TAPAS [17] and XONN [32] compressed models into binary neural network formats, while studies like CryptoNAS [30], Sphynx [33], and SafeNet [34] used Neural Architecture Search (NAS) to find optimal architectures. Delphi [29], SNL [35], and DeepReduce [36] replaced ReLU functions with low order polynomial or linear functions to accelerate MPC-based Private Inference (PI). Nevertheless, these strategies proved suboptimal for homomorphic encrypted inference. Solutions like LoLa [20], CHET [37], and HEAR [38] sought to use ciphertext packing techniques but were not optimized for GCN-based models.

**Threat Model.** In line with other studies [12, 37, 39], we presume a semi-honest cloud-based machine learning service provider. The well-trained STGCN model's weight, including adjacency matrices, is encoded as plaintexts in the HE-inference process. Clients encrypt data via HE and upload it to the cloud for privacy-preserving inference service. The server completes encrypted inference without data decryption or accessing the client's private key. Finally, clients decrypt the returned encrypted inference results from the cloud using their private keys.

## 3 LinGCN Framework

### 3.1 Motivation

We present two pivotal observations that serve as the foundation for our LinGCN.

**CKKS Based Encrypted Multiplication.** As described in section 2, the message $m$ is scaled by a factor $\Delta = 2^p$ before encoding, resulting in $\Delta m$. Any multiplicative operations square the ciphertext's scale, including inherent noise $e$. The $Rescale$ operation reduces this scale back to $\Delta$ by modswitch [15], reducing $Q$ by $p$ bits and lowering the level, as depicted in Figure 2. When $Q \to q_o$ after $L$ multiplications and rescaling, the ciphertext's multiplicative level reaches 0. At this stage, bootstrapping [12] is necessary for further multiplication, unless a higher $Q$ is used to increase the multiplicative level $L$.

**Observation 1: Saving Level is Important in CKKS Scheme.** Within a STGCN model, the graph embedding is encrypted with an initial polynomial degree, denoted as $N$, and a coefficient modulus,

represented by $Q$. These factors are determined by the network's multiplicative depth. As the multiplication depth increases, the coefficient modulus must also increase, subsequently leading to a higher polynomial degree $N$. To maintain a security level above a certain threshold (for example, 128-bit) in the face of deeper multiplicative depths, a larger $N$ is necessitated [37, 38, 12]. A larger $N$ inherently results in increased complexity in each homomorphic encryption operation such as Rot and CMult.

These operations are integral to the computation of convolution (Conv) and graph convolution (GCN-Conv). We present a comparative analysis of operator latency with differing polynomial degrees $N$ in Figure 2. Thus, it can be inferred that pruning certain operators to conserve levels not only diminishes the latency of the pruned operators but also reduces other operators' latency.

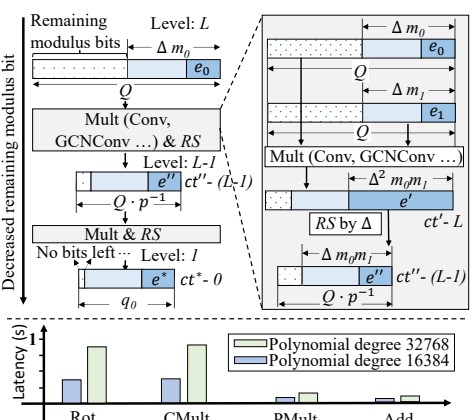

**Observation 2: Synchronized Linearization Matters.** Our study mainly targets reducing multiplication depth in standard STGCN layers, comprising an initial spatial GCNConv operation, two non-linear operators, and a temporal convolution. Under the CKKS base HE scheme, each operator's input needs synchronized multiplication depths, with any misalignment adjusted to the minimal depth. This is particularly vital for the GCNConv operator, while later node-wise separable operators enable structured non-linear reduction to boost computational efficiency.

Figure 2: Top: Rescale decreases the ciphertext level. Bottom: Higher polynomial degree leads to longer HE operator's latency.

Unstructured non-linear reduction strategies, common in MPC setups like Auto-ReP [40], PASNet [41] SNL [35], RR-Net [42], DELPHI [29], and SAFENet [43], prove suboptimal for homomorphic encrypted inference. Our example in Figure 3(b) illustrates node-wise unstructured non-linear reduction in an STGCN layer, leading to unsynchronized remaining multiplication depth budgets for nodes after the layer. This approach is ineffective in reducing levels consumption for homomorphic encrypted inference, emphasizing the necessity for a more structured non-linear

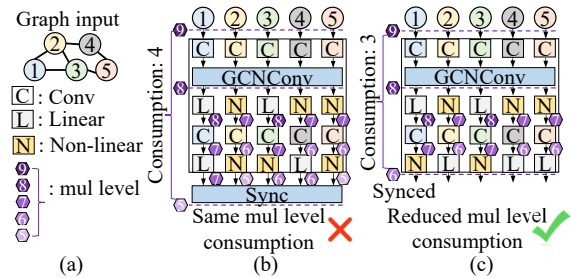

Figure 3: Unstructured vs. structural linearization. Unstructured one doesn't lead to effective level reduction.

reduction approach. To address the limitations of unstructured non-linear reduction, we propose a structured approach, illustrated in Figure 3(c), where we apply an equal level of non-linear reduction to each target node, reducing the overall multiplication depth. In contrast to CryptoGCN's layer-wise pruning method [12], our scheme provides nodes with the flexibility to perform non-linear operations at preferred positions. This fine-grained, structural non-linear reduction may improve the non-linear reduced structure, potentially augmenting the efficiency and effectiveness of HE inference.

## 3.2 Learning for Structural Linearization

**Problem Formulation.** Our methodology is aimed at the node-wise structural dropping of non-linear operators within a given $L$-layered STGCN model $f_W$, parameterized by $W = \{W_i\}_{i=0}^{2L}$. This model maps the input $X_0 \in R^{V \times C \times T}$ to the target $Y \in R^d$. Here, $V$, $C$, and $T$ represent the number of nodes, channels, and frames, respectively. The primary objective of our approach is to eliminate the non-linear operator (designated as $\sigma_n$) in a structured, node-wise manner. The ultimate aim is to achieve a reduced multiplication level while minimizing any resultant drop in model accuracy. This goal serves to optimize the efficiency of homomorphic encrypted inference for STGCN model.

We introduce an indicator parameter, denoted as $h$, to mark the node-wise non-linear operator dropping. The following conditions apply: (1) If $h_{i,k} = 0$, the non-linear operator $\sigma_n$ is substituted by the linear function $f(x) = x$; (2) If $h_{i,k} = 1$, the operator $\sigma_n$ is utilized. Here, $h_{i,k}$ signifies

the $k_{th}$ node of the $i_{th}$ non-linear layer. The proposed node-wise non-linear indicator parameter $h_{i,k}$ facilitates the expression of the $i_{th}$ non-linear layer with partial linearization, as given by $X_{i,k} = h_{i,k} \odot \sigma_n(Z_{(i-1),k}) + (1 - h_{i,k}) \odot Z_{(i-1),k}$. Here, $Z_{i-1}$ denotes the input of the $i_{th}$ non-linear layer. Consequently, the issue of structural linearization can be formulated as follows:

$$\underset{W,h}{\operatorname{argmin}} \mathcal{L} = \underset{W,h}{\operatorname{argmin}} \quad \mathcal{L}_{acc}(f_W(X_0), Y) + \mu \cdot \sum_{i=1}^{2L} ||h_i||_0 \tag{2}$$
$$\text{subject to } \forall j, k \in [1, V], \ (h_{2i,j} + h_{2i+1,j}) = (h_{2i,k} + h_{2i+1,k})$$

In our formulation, $||h_r||0$ represents the $L_0$ norm or the count of remaining non-linear operators. $\mathcal{L}acc$ signifies cross-entropy loss, while $\mu$ is the $L_0$ norm penalty term in the linearization process. The second term of Eq. 2 poses a challenge due to its non-differentiability, stemming from zero norm regularization and additional constraints on the discrete indicator parameter $h_i$. Hence, this issue becomes difficult to handle using traditional gradient-based methods.

**Differentiable Structural Linearization.** To handle the non-differentiable discrete indicator parameter, we introduce a trainable auxiliary parameter, $h_w$, to denote its relative importance. However, the transformation from $h_w$ to the final indicator $h$ must comply with the structural constraint in Eq.2, ruling out simple threshold checks used in SNL [35]. Structural pruning methods [44, 45], useful for weight or neuron pruning, are ill-suited for our problem due to its exponential complexity arising from 2 non-linear selections of 25 nodes, and its distinct nature from weight or neuron pruning.

To tackle this challenge, we propose a structural polarization forward process, detailed in Algorithm 1. The algorithm first ranks the relative importance of two auxiliary parameters ($h_{w(2i,j)}$ and $h_{w(2i+1,j)}$) for every $j_{th}$ node within the $i_{th}$ STGCN layer, obtaining the higher and lower values and their respective indices. We then sum each $i_{th}$ layer's higher auxiliary parameter value into $s_h$, and lower auxiliary parameter value into $s_l$. Next, we conduct a threshold check of $s_h$ and assign the final polarization output of the indicator value into the corresponding locations of higher value indices, applying the same polarization for lower values. The proposed structural polarization enforces the exact constraint as given in Eq. 2, where each STGCN layer maintains a synchronized non-linear count across all nodes. The structural polarization is also

---

**Algorithm 1** Structural Polarization.

**Input:** $h_w$: auxiliary parameter
**Output:** $h$: final indicator
1: **for** $i = 0$ to $L$ **do**
2:     $s_h, s_l = 0$ and $ind_h, ind_l \leftarrow \emptyset$
3:     **for** $j = 1$ to $V$ **do**
4:         **if** $h_{w(2i,j)} > h_{w(2i+1,j)}$ **then**
5:             $s_h += h_{w(2i,j)}, s_l += h_{w(2i+1,j)}$
6:             $ind_h \leftarrow (2i, j), ind_l \leftarrow (2i + 1, j)$
7:         **else**
8:             $s_h += h_{w(2i+1,j)}, s_l += h_{w(2i,j)}$
9:             $ind_h \leftarrow (2i + 1, j), ind_l \leftarrow (2i, j)$
10:         **end if**
11:     **end for**
12:     $h_{ind_h} = s_h > 0$ and $h_{ind_l} = s_l > 0$
13: **end for**

---

capable of capturing the relative importance of the auxiliary parameter within each node and applying the proper polarization to every position, all with a complexity of only $O(V)$. Each STGCN layer retains the flexibility to choose all positions to conduct the non-linear operation, or just select one position for each node to conduct the non-linear operation, or opt not to perform the non-linear operation at all. This selection is based on the importance of their auxiliary parameters.

Despite its benefits, the structural binarization in Algorithm 1 is discontinuous and non-differentiable. To approximate its gradient, we use coarse gradients [46] via straight through estimation (STE), a good fit for updating $h_w$ and reflecting the optimization objective in Eq.2. Among various methods such as Linear STE[47], ReLU and clip ReLU STE [48], we adopt Softplus STE [49] due to its smoothness and recoverability discovered in [49], to update $h_w$ as follows:

$$\frac{\partial \mathcal{L}}{\partial h_{w(i,k)}} = \frac{\partial \mathcal{L}_{acc}}{\partial X_{i,k}}(\sigma_n(Z_{i-1}) - Z_{i-1})\frac{\partial h_{i,k}}{\partial h_{w(i,k)}} + \mu\frac{\partial h_{i,k}}{\partial h_{w,(i,k)}}, \quad \frac{\partial h_{i,k}}{\partial h_{w(i,k)}} = Softplus(h_{w(i,k)}) \tag{3}$$

The gradient of the auxiliary parameter, as displayed in Eq. 3, consists of two components: the gradient from the accuracy constraint and the gradient from the linearization ratio constraint. The gradient stemming from the accuracy constraint will be negative if it attempts to counteract the linearization process, thereby achieving a balance with the $L_0$ penalty term $\mu \cdot \sum_{i=1}^{2L} ||h_i||_0$. The

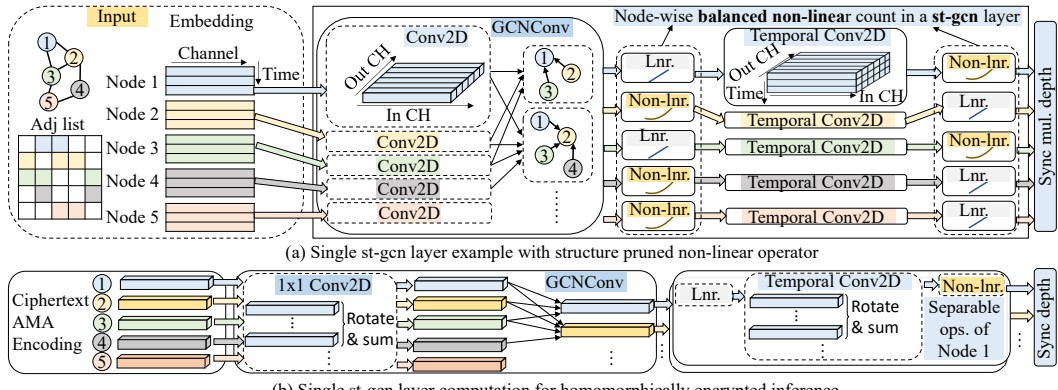

(a) Single st-gcn layer example with structure pruned non-linear operator

(b) Single st-gcn layer computation for homomorphically encrypted inference

Figure 4: Overview of single layer STGCN with non-linear reduction and computation.

entirety of the linearization process is recoverable, which allows for the exploration of a near-optimal balance between accuracy and linearization ratio via the back-propagation process.

### 3.3 Learnable Polynomial Replacement with Teacher Guidance

CryptoNet [19] and DELPHI [29] suggest using quadratic functions $y = x^2$ as ReLU replacements for private inference. A second-order polynomial replacement [50] $y = x^2 + x$ for ReLU was proposed but resulted in decreased performance. CryptoGCN [12] uses a layer-wise trainable polynomial function $(ax^2 + bx + c)$ for the STGCN model, but it suffers from significant accuracy loss and scalability problems. To tackle these issues, we offer a two-tiered strategy: node-wise trainable polynomial replacement and a distillation technique to facilitate the convergence of replacement.

**Node-wise Trainable Polynomial Replacement.** In our HE setup (see Figure 4), each node can independently perform non-linear, convolution, and further non-linear operations without increasing the multiplication depth. We suggest using a node-wise trainable polynomial function as the non-linear function (see Eq. 4). To prevent gradient explosion from quadratic terms, we include a small constant $c$ to adjust the gradient scale of the $w_2$ parameter [41].

$$\sigma_n(x) = c \cdot w_2 x^2 + w_1 x + b \tag{4}$$

**Improved Convergence Through Distillation.** Training a polynomial neural network from scratch often overfits, leading to accuracy loss [50, 12]. To counteract this, we recommend initially training a baseline ReLU model as the teacher, then transferring its knowledge to the polynomial student model to lessen accuracy degradation. The student model's parameters are initialized from the teacher's and polynomial function parameters start at $(w_2 = 0, w_1 = 1, b = 0)$. To aid knowledge transfer, we use KL-divergence loss [51] and peer-wise normalized feature map difference penalty. The loss function for polynomial replacement is as follows:

$$\mathcal{L}_p = (1 - \eta)\mathcal{L}_{CE}(f_{W,s}(X_0), Y) + \eta\mathcal{L}_{KL}(f_{W,s}(X_0), f_{W,t}(X_0)) + \frac{\varphi}{2}\sum_{i=1}^{L} MSE(\frac{X_{i,s}}{||X_{i,s}||_2}, \frac{X_{i,t}}{||X_{i,t}||_2}) \tag{5}$$

In the above equation, $\eta$ is employed to balance the significance between the CE loss and the KL divergence loss components, while $\varphi$ acts as the penalty weight for the peer-wise feature map normalized $L_2$ distance term, analogous to the approach in [52].

### 3.4 Put it Together

The comprehensive workflow of our proposed LinGCN framework is outlined in Algorithm 2. Initially, we construct the student model, $M_S$, utilizing the parameters of the teacher model, $M_T$, and subsequently initialize the indicator auxiliary parameter $h_w$ for $M_S$. Following this, $M_S$ undergoes training iterations involving updates for structural linearization. Upon

achieving convergence, the indicator function $h$ is fixed, and the ReLU function in $M_S$ is replaced with a polynomial function. Subsequently, we initialize $w_{poly}$ and train the final model using a combination of CE loss and the two-level distillation loss as defined in Eq. 5.

The resulting model output will feature structurally pruned node-wise polynomial functions, rendering it primed for the final stage of homomorphic encryption-based inference.

**Further Operator Fusion to Save Level.** Figure 4 shows an example of the single-layer STGCN structure generated by our LinGCN framework. The STGCN utilizes both input nodes information and aggregates them to generate output node features. We adopt the same AMA format [12] for GCNConv computation, as such, we are able to fuse the plaintext weights $c \cdot w_2$ of node-wise polynomial function into GCNConv layer and save one multiplication depth budget. Layer operators involve another temporal convolution layer and non-linear/linear layer, we can still fuse the plaintext weights $c \cdot w_2$ from the second polynomial function into the convolution operator and save the multiplication depth budget. As such, the ReLU-reduced example shown in Figure 4 only consumes 3 multiplication level of ciphertext as opposed to the original 4 multiplication depth budget. For the convolution operator, we adopt the same computation flow from [12, 38].

---

**Algorithm 2** LinGCN Framework Workflow.

**Input:** Pretrain ReLU-based model $M_T$, linearization penalty $\mu$ and optim. $OP_L$, polynomial replacement optim. $OP_P$
**Output:** Level-reduced polynomial model
1: Copy $M_S$ from $M_T$
2: Initialize $h_w$ for $M_S$
3: **for** Structural linearization iterations **do**
4:     Calculate $\mathcal{L}$ via Eq. 2 and Algorithm 1
5:     Update $W$ and $h_w$ through back propagation (Eq. 3) by minimizing $\mathcal{L}$ using $OP_L$
6: **end for**
7: Freeze $h_w$ and $h$
8: Replace ReLU in $M_s$ with polynomial
9: Initialize $w_{poly}$
10: **for** Polynomial replacement iterations **do**
11:     Calculate $\mathcal{L}_p$ via Eq. 5
12:     Update $W$ through back propagation by minimizing $\mathcal{L}_p$ using $OP_P$
13: **end for**

---

## 4 Experiment

### 4.1 Experiment Setting

**HE Parameter Setting.** We performed experiments with two sets of encryption parameters: one without and one with non-linear reduction. We used a scaling factor $\Delta = 2^{33}$ in both cases to ensure comparable accuracy. This scaling consumes 33 bits from the ciphertext modulus $Q$ per level. Without non-linear reduction, a 3-layer STGCN network needs 14 levels, and a 6-layer network needs 27. For a security level of 128-bit, we set $Q$ to 509 (932) and the polynomial degree $N$ to $2^{15}(2^{16})$ for 3 (6) layers STGCN network. In the non-linear reduction case, the total level count ranges from 14 to 9 for 3-layer, and 27 to 16 for 6-layer networks. With the reduced models, we can use smaller Q and $N$ while maintaining 128-bit security. Specifically, we set $Q$ to 410 to 344 and $N$ to $2^{14}$ for the 3-layer network (from 3 non-linear reduced), and $Q$ to 767 to 569 and $N$ to $2^{15}$ for the 6-layer network (starting from 5 non-linear reduced).

**Experiment Dataset.** The **NTU-RGB+D** dataset [21], the largest available with 3D joint annotations for human action recognition, is utilized in our study. With 56,880 action clips in 60 classes, each is annotated with $(X, Y, Z)$ coordinates of 25 joints (nodes) per subject. We chose the NTU-cross-View (NTU-XView) benchmark given its representativeness as a human skeleton joint dataset, containing 37,920 training and 18,960 evaluation clips. For a thorough evaluation, 256 frames were used from each clip, resulting in a $2 \times 3 \times 256 \times 25$ input tensor for two individuals each with 3 channels.

**Baseline Model.** Our experiment runs on 8*A100 GPU server, and uses the state-of-the-art STGCN architecture for human action recognition, which combines GCN and CNN. We tested three configurations: STGCN-3-128, STGCN-3-256, and STGCN-6-256, where the first number represents the layer count, and the second the last STGCN layer's channel count. These networks contain a stack of three or six STGCN layers, followed by a global average pooling layer and a fully-connected layer. They were trained for 80 epochs using SGD, with a mini-batch size of 64, a momentum of 0.9, weight decay of $10^{-4}$, and dropout 0.5. The initial learning rate (LR) was 0.1, with a decay factor of 0.1 at the 10th and 50th epochs. The baseline accuracy of models can also be found in Table 1.

**LinGCN Algorithm Setting.** In our LinGCN algorithm, we used the baseline model from Table 1 as the teacher model in Algorithm 2. For structural linearization, we used SGD optimizer with learning rate (LR) 0.01 for both weight and auxiliary parameters. The $L_0$ norm penalty $\mu$ was varied from 0.1

Table 1: All ReLU based model architecture and accuracy

| Model | Layer-wise number of channel configuration | Accuracy (%) |
|---|---|---|
| STGCN-3-128 | 3-64-128-128 | 80.64 |
| STGCN-3-256 | 3-128-256-256 | 82.80 |
| STGCN-6-256 | 3-64-64-128-128-256-256 | 84.52 |

to 10 for desired linearized layers, over 25 epochs. Then, ReLU was replaced with a second-order polynomial in the polynomial replacement iterations. We set scaling factor $c$ at 0.01, parameters $\eta$ and $\varphi$ at 0.2 and 200 respectively. This process took 90 epochs, with SGD optimizer and initial LR of 0.01, decaying by a factor of 0.1 at the $40th$ and $80th$ epochs.

**Private Inference Setup.** Our experiments used an AMD Ryzen Threadripper PRO 3975WX machine, single threaded. The RNS-variant of the CKKS scheme [53] was employed using Microsoft SEAL version 3.7.2 [54]. We adopted the Adjacency Matrix-Aware (AMA) data formatting from CryptoGCN [12] to pack input graph data and performed GCNConv layer inference via multiplying with sparse matrices $A_i$ split from the adjacency matrix $A$. The temporal convolution operation was optimized using the effective convolution algorithm from [38, 12, 55]. The computation for the global average pooling layer and fully-connected layer also follows the SOTA [38, 12].

<table>
<tr><th colspan="4">Table 2: STGCN-3-128 comparison</th><th colspan="4">Table 3: STGCN-3-256 comparison</th></tr>
<tr><th>Model</th><th colspan="3">STGCN-3-128</th><th>Model</th><th colspan="3">STGCN-3-256</th></tr>
<tr><th>Methods</th><th>Non-linear layers</th><th>Test acc (%)</th><th>Latency (s)</th><th>Methods</th><th>Non-linear layers</th><th>Test acc (%)</th><th>Latency (s)</th></tr>
<tr><td>LinGCN</td><td>6</td><td>77.55</td><td>1856.95</td><td>LinGCN</td><td>6</td><td>80.29</td><td>4632.05</td></tr>
<tr><td>LinGCN</td><td>5</td><td>75.48</td><td>1663.13</td><td>LinGCN</td><td>5</td><td>79.07</td><td>4166.12</td></tr>
<tr><td>LinGCN</td><td>4</td><td>76.33</td><td>1458.95</td><td>LinGCN</td><td>4</td><td>78.59</td><td>3699.49</td></tr>
<tr><td>LinGCN</td><td>3</td><td>74.27</td><td>850.22</td><td>LinGCN</td><td>3</td><td>76.41</td><td>2428.88</td></tr>
<tr><td>LinGCN</td><td>2</td><td>75.16</td><td>741.55</td><td>LinGCN</td><td>2</td><td>74.74</td><td>2143.46</td></tr>
<tr><td>LinGCN</td><td>1</td><td>69.61</td><td>642.06</td><td>LinGCN</td><td>1</td><td>71.98</td><td>1873.40</td></tr>
<tr><td>CryptoGCN</td><td>6</td><td>74.25</td><td>4273.89</td><td>CryptoGCN</td><td>6</td><td>75.31</td><td>10580.41</td></tr>
<tr><td>CryptoGCN</td><td>5</td><td>73.12</td><td>1863.95</td><td>CryptoGCN</td><td>5</td><td>73.78</td><td>4850.93</td></tr>
<tr><td>CryptoGCN</td><td>4</td><td>70.21</td><td>1856.36</td><td>CryptoGCN</td><td>4</td><td>71.36</td><td>4831.93</td></tr>
</table>

## 4.2 Experiment Result and Comparison

**LinGCN Outperforms CryptoGCN [12].** The proposed **LinGCN** algorithm demonstrates superior performance compared to CryptoGCN [12]. The STGCN-3-128 and STGCN-3-256 models, which are the basis of this comparison, share the same backbone as those evaluated in CryptoGCN [12]. Hence, we have undertaken a comprehensive cross-work comparison between the LinGCN framework and CryptoGCN [12]. The latter employs a heuristic algorithm for layer-wise activation layer pruning, which necessitates a sensitivity ranking across all layers. This approach, however, proves to be ineffective, leading to significant accuracy drops as the number of pruned activation layers increases. Comparative results between LinGCN and CryptoGCN [12] are provided in Table 2 and Table 3. In the context of LinGCN, the number of non-linear layers presented in the table represents the effective non-linear layers post-structural linearization. For the STGCN-3-128 and STGCN-3-256 models, LinGCN yielded an accuracy of 77.55% and 80.28% respectively for the baseline 6 non-linear layers model, exceeding CryptoGCN [12] by 3.3% and 4.98%. This outcome signifies the superiority of our proposed teacher-guided node-wise polynomial replacement policy over the naive layer-wise polynomial replacement provided in CryptoGCN [12].

Importantly, the LinGCN model for the 6 non-linear layers employs a more fine-grained node-wise polynomial function, which is fused into the preceding Conv or GCNConv layers. Thus, the encryption level required is lower than that of CryptoGCN [12], resulting in reduced latency. When considering non-linear layer reduction performance, LinGCN experiences only a 1.2 to 1.7 % accuracy decline with a 2 effective non-linear layer reduction, whereas CryptoGCN [12] displays approximately 4% accuracy degradation when pruned for 2 non-linear layers. Remarkably, LinGCN achieves an accuracy improvement of more than 6% over CryptoGCN [12] for models with 4 non-linear layers. Even in models with only 2 non-linear layers, LinGCN exhibits exceptional accuracy

(75.16% and 74.74%). While this accuracy aligns with the performance of 6 non-linear layer models in CryptoGCN [12], LinGCN significantly outperforms it in terms of private inference latency, providing more than a five-fold reduction compared to CryptoGCN [12].

**Non-linear Layer Sensitivity.** We conduct a sensitivity analysis on the STGCN-3-256 model to investigate the effect of non-linear layers. The results are displayed in Figure 5, wherein the $i$ layers in the figure represent the remaining effective non-linear layers in the model, and every $2i-1$ and $2i$ exhibit a total number of non-linear nodes summing up to an integer multiple of 25. During the automatic structural linearization process, the gradient propagation is orchestrated to

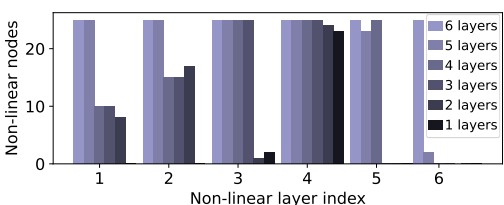

Figure 5: STGCN-3-256 structural linearization

strike a balance between two goals: (i) preserving the feature structural information at the deeper layers, and (ii) mitigating the over-smoothing effect [56] resulting from the deep linearized GCN layers. Consequently, the middle 4th layer's nonlinearity is most important within the STGCN model.

**LinGCN is Scalable.** Beyond the evaluation of the 3-layer model, we also apply our LinGCN to a 6-layer STGCN-6-256 model, which no prior work evaluated, maintaining the same settings to assess the trade-off between accuracy and latency. An intriguing finding is that the teacher-guided full polynomial replacement not only maintains model performance but also surpasses the accuracy of the teacher baseline by 0.95%, achieving an accuracy of 85.47%. This suggests that the STGCN model equipped with a smooth polynomial activation function may enhance model expressivity. Experiments reveal a significant redundancy in the non-linear layers of the STGCN-6-256 model, as it exhibits no

Table 4: LinGCN for STGCN-6-256 model.

| Model | STGCN-6-256 | | |
|---|---|---|---|
| Methods | Non-linear layers | Test acc (%) | Latency (s) |
| LinGCN | 12 | 85.47 | 21171.80 |
| LinGCN | 11 | 86.24 | 19553.96 |
| LinGCN | 7 | 85.08 | 8186.35 |
| LinGCN | 5 | 83.64 | 7063.51 |
| **LinGCN** | **4** | **85.78** | **6371.39** |
| LinGCN | 3 | 84.28 | 5944.81 |
| LinGCN | 2 | 82.27 | 5456.12 |
| LinGCN | 1 | 75.93 | 4927.26 |

accuracy degradation up to the reduction of 8 non-linear layers. When the pruning extends to 10 non-linear layers, the accuracy experiences a slight dip of 3.2% compared to the non-reduction one.

**Pareto Frontier of LinGCN.** Figure 1 compares the Pareto frontier between LinGCN and CryptoGCN [12], showing LinGCN consistently surpassing CryptoGCN by at least 4% in accuracy across all latency ranges. As latency constraints ease (around 6000s), LinGCN's accuracy notably increases, improving by roughly 10% over CryptoGCN. Impressively, LinGCN maintains an accuracy of 75.16% while achieving a private inference latency of 742s, making it 14.2 times faster than CryptoGCN.

### 4.3 Abalation Studies

**Non-linear Reduction & Replacement Sequence matters.** Our LinGCN algorithm employs a sequence of non-linear reduction and polynomial replacement, proving more efficacious in terms of search space convergence compared to the inverse sequence of polynomial replacement followed by non-linear reduction. The reasoning behind this lies in the possibility that the polynomial replacement process may produce a model optimized for a given architecture, hence subsequent changes to the architecture, such as non-linear reduction, could lead to significant accuracy degradation. Keeping all other parameters constant and merely altering the replacement sequence, we present the accuracy evaluation results of STGCN-3-256 model in Figure 6a. As depicted in the figure, the sequence of polynomial replacement followed by non-linear reduction incurs an accuracy degradation exceeding 2% compared to the baseline sequence across all effective non-linear layer ranges.

**LinGCN Outperforms Layer-wise Non-linear Reduction.** A significant innovation in our LinGCN framework is the structural linearization process. This process allows nodes to determine their preferred positions for non-linearities within a single STGCN layer, as opposed to enforcing a rigorous constraint to excise the non-linearity across all nodes. This less restrictive approach yields a more detailed replacement outcome compared to the layer-wise non-linear reduction. To demonstrate the impact of this method, we kept all other parameters constant for of STGCN-3-256 model, altering

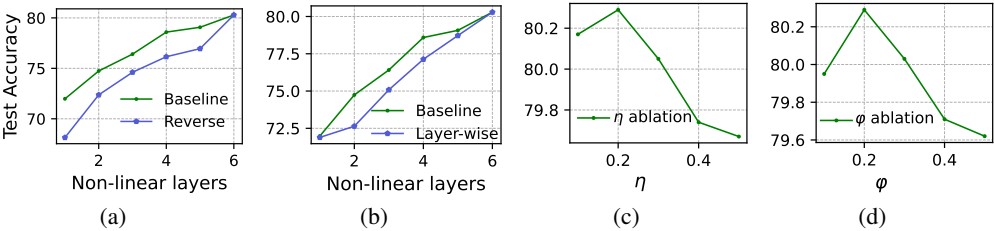

Figure 6: Ablation study for (a) Replacement sequence, (b) node-wise vs. layer-wise linearization. (c) $\eta$: KL divergence distillation parameter (d) $\varphi$: feature map distance distillation parameter.

only the structural linearization to layer-wise linearization, and subsequently evaluated the model's accuracy, as shown in Figure 6b. The results indicate that layer-wise linearization results in a 1.5% decrease in accuracy compared to structural linearization within the range of 2 to 5 effective non-linear layers. This demonstrates the effectiveness of the proposed structural linearization methodology.

**Distillation Hyperparameter Study.** In prior experiments, we kept $\eta$ and $\varphi$ at 0.2 and 200 respectively. To assess their influence during distillation, we tested $\eta$ and $\varphi$ across a range of values while distilling the STGCN-3-256 model with 6 non-linear layers. We test $\eta \in [0.1, 0.2, 0.3, 0.4, 0.5]$ and $\varphi \in [100, 200, 300, 400, 500]$. Notably, we maintained $\varphi = 200$ and $\eta = 0.2$ for the respective $\eta$ and $\varphi$ ablations. The study, shown in Figure 6c and Figure 6d, confirms that $\eta = 0.2$ and $\varphi = 200$ yield optimal accuracy, and larger penalties might lower accuracy due to mismatch with the teacher model. This study thus justifies our selection of the $\eta$ and $\varphi$.

**LinGCN Generalize to Other Dataset.** Without loss of generality, we extended our evaluation on Flickr [57] dataset, which is a representative node classification dataset widely used in GNN tasks.It consists of 89,250 nodes, 989,006 edges, and 500 feature dimensions. This dataset's task involves categorizing images based on descriptions and common online properties. For the security setting, we assume that node features are user-owned and the graph adjacency list is public. The Flickr dataset has a larger adjacent list but smaller feature dimension compared to the NTU-XVIEW dataset. We utilize three GCN layers with 256 hidden dimensions. Each GCN layer has similar structure as STGCN backbone architecture and has two linear and nonlinear layers. We conduct full-batch GCN training to obtain ReLU-based baseline model accuracies of 0.5492/0.5521 for validation/test dataset. We obtain the accuracy/latency tradeoff detailed in the Table 5. LinGCN framework substantially diminishes the number of effective nonlinear layers, which leads to 1.7 times speedup without much accuracy loss.

Table 5: LinGCN for Flickr dataset.

| Num. of nonlinear layers in GCN layers | Accuracy (val/test, %) | Latency (s) |
|---|---|---|
| 6 | 0.5281/0.5275 | 4290.93 |
| 2 | 0.5247/0.5266 | 2740.94 |
| 1 | 0.5269/0.5283 | 2525.80 |

## 5  Discussion and Conclusion

Our LinGCN optimizes HE-based GCN inference by reducing multiplication levels through a differentiable structural linearization algorithm and a compact node-wise polynomial replacement policy, both guided by a two-level distillation from an all-ReLU teacher model. Additionally, we improve HE solutions for GCN private inference, enabling finer operator fusion for node-wise activation functions. LinGCN outperforms CryptoGCN by 5% in accuracy and reducing latency by 14.2 times at ~75% accuracy, LinGCN sets a new state-of-the-art performance for private STGCN model inference. In the future, we will expand the proposed algorithm on other neural networks.

## Acknowledgement

This work was in part supported by the NSF Grants CNS-2348733, 2247892, 2247893, OAC-2319962, Semiconductor Research Corporation (SRC) Artificial Intelligence Hardware program, and UConn CACC center. Any opinions, findings and conclusions, or recommendations expressed in this material are those of the authors and do not necessarily reflect the views of the funding agencies.

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

# A  Appendix

## A.1  HE encoding with AMA format

Prior to encoding input data into polynomials, it is necessary to map the four-dimensional tensor $X \in R^{B \times C \times T \times J}$ to a one-dimensional vector in $R^{N/2}$ using the AMA format, as proposed in [12]. This transformation allows for more efficient execution of STGCN forward-computation in the HE domain. Below, we present the definition of the $Vec$ function employed to map tensor $X$ to a vector in $R^{N/2}$:

$$Vec(X) = y_j = (y_{0,j}, \dots, y_{i,j}, \dots, y_{N/2,j}) \in R^{N/2}$$
$$s.t.\ y_{i,j} = X_{((i\,mod\,T)\%B) \times (i\,mod\,B \cdot T) \times (i\,\%\,T) \times j} \quad (6)$$
$$j \in J$$

Following the mapping process, the vectors $y_j$ are encoded into polynomials with degree $N$ and subsequently encrypted into ciphertext $ct_j$, as detailed in [53]. In this study, when N is set to $2^{16}$, all tensors, including intermediate tensors, can be encrypted and packed into 25 ciphertexts, which corresponds to the number of nodes. For cases where $N = 2^{15}$ ($2^{14}$) the number of ciphertexts is 50(100). By selecting an appropriate value for N, the encryption and packing processes can be optimized to maintain performance and efficiency.

## A.2  HE Setting Details

In Table 6, we furnish comprehensive details regarding the HE inference parameters. Specifically, i-STGCN-3 denotes a 3-layer STGCN model with i effective non-linear layers, while i-STGCN-6 signifies a 6-layer STGCN model with i effective non-linear layers. In this context, N represents the polynomial degree, and Q corresponds to the coefficient modulus.

To guarantee computation precision utilizing a one-time rescale operation, we assign the scale factor $p$ for both ciphertext and plaintext to $2^{33}$. This allocation results in a reduction of the current Q of ciphertext by p bits. This setup ensures that the overall performance and accuracy conform to the desired criteria while capitalizing on the security and resilience advantages conferred by HE.

Table 6: HE parameter settings in detail.

| Model | Encryption Parameters | | | | Mult Level |
|---|---|---|---|---|---|
| | N | Q | p | $q_0$ | |
| 6-STGCN-3 | 32768 | 509 | 33 | 47 | 14 |
| 5-STGCN-3 | 32768 | 476 | 33 | 47 | 13 |
| 4-STGCN-3 | 32768 | 443 | 33 | 47 | 12 |
| 3-STGCN-3 | 16384 | 410 | 33 | 47 | 11 |
| 2-STGCN-3 | 16384 | 377 | 33 | 47 | 10 |
| 1-STGCN-3 | 16384 | 344 | 33 | 47 | 9 |
| 12-STGCN-6 | 65536 | 932 | 33 | 41 | 27 |
| 11-STGCN-6 | 65536 | 899 | 33 | 41 | 26 |
| 7-STGCN-6 | 32768 | 767 | 33 | 41 | 22 |
| 5-STGCN-6 | 32768 | 701 | 33 | 41 | 20 |
| 4-STGCN-6 | 32768 | 668 | 33 | 41 | 19 |
| 3-STGCN-6 | 32768 | 635 | 33 | 41 | 18 |
| 2-STGCN-6 | 32768 | 602 | 33 | 41 | 17 |
| 1-STGCN-6 | 32768 | 569 | 33 | 41 | 16 |

## A.3  HE inference on GCNConv and Temporal-Conv Layer

Upon obtaining the AMA-packed ciphertexts $ct_j$, the adjacency matrix multiplication $A \cdot X$ can be decomposed into a series of plaintext multiplications, *PMult*, in the HE domain. This decomposition accelerates HE-inference without necessitating rotations. Furthermore, the subsequent temporal convolution is performed solely on the temporal dimension $T$, utilizing $1 \times 9$ kernels.

Table 7: Comparison of latency breakdown between the non-reduced model with optimized model.

| Model | HE Operators latency (s) | | | | Total Latency | Speedup |
|---|---|---|---|---|---|---|
| | Rot | PMult | Add | CMult | (s) | (×) |
| 6-STGCN-3-128 | 1336.25 | 378.25 | 99.65 | 37.45 | 1851.60 | - |
| 2-STGCN-3-128 | 392.21 | 266.13 | 68.90 | 14.31 | 741.55 | 2.50 |
| 6-STGCN-3-256 | 2641.09 | 1508.19 | 397.17 | 74.90 | 4621.36 | - |
| 2-STGCN-3-256 | 777.68 | 1062.21 | 274.96 | 28.63 | 2143.47 | 2.16 |
| 12-STGCN-6-256 | 18955.09 | 1545.09 | 396.23 | 275.39 | 21171.80 | - |
| 2-STGCN-6-256 | 4090.08 | 1006.79 | 244.19 | 115.05 | 5456.12 | 3.88 |

$$ct'_k = A \cdot X = \sum_{i=1}^{m} ct_k A_i = \sum_{i=1}^{m} \sum_{k=1}^{J} PMult(ct_{i_k}, a_{i_k k}) \tag{7}$$

The AMA-packed ciphertexts allow for natural temporal convolution by single-node ciphertext $ct_j$, facilitating independent computation. This approach results in a ReLU-reduction design through structural pruning of ReLUs. The primary constraint to consider in this context is ensuring that the level consumption of each ciphertext remains equal prior to the GCNConv layer (node aggregation).

### A.4 Further Detail of Operator Fusion

During the HE-inference process, employing weight fusion conserves the multiplicative depth, consequently reducing the ciphertext level budget. For instance, batch normalization, defined by an affine transformation $a'x + b'$, and a polynomial activation function, defined by $(ax + b)^2 + c$, can be readily fused into the corresponding temporal convolution layer $wx + b''$ with the function $w(a(a'x + b') + b) + b'' = (w \cdot a \cdot a')x + ab' + wb + b''$. As a result, three consecutive multiplications are consolidated into a single multiplication (pre-computing $w \cdot a \cdot a'$), thereby reducing the level consumption of ciphertext from 4 to 2 ($1 \times 9$ convolution, batch normalization, and polynomial activation).

Analogous to the temporal-convolutional layer, the same fusion strategies can be applied to the polynomial activation and batch normalization of the GCNConv layer. Utilizing AMA-packed ciphertexts, the node aggregation in GCNConv is translated as depicted in Equation 7, where each ciphertext carries out scalar multiplication with the plaintext of matrix elements $a_{i_k,k}$. Consequently, these plaintext matrix elements $a_{i_k,k}$ are fused with the primary $1 \times 1$ convolutional kernels to conserve the multiplicative level, reducing the total level consumption of the GCNConv layer from 4 to 2 ($1 \times 1$ convolution, adjacency matrix multiplication, batch normalization, and polynomial activation).

### A.5 Operator latency breakdown

Table 7 presents a comprehensive operator latency breakdown encompassing Rot, PMult, Add, and CMult operations. The designation i-STGCN-3-128 refers to an STGCN-3-128 model with i residual non-linear layers. As indicated in the table, the non-linear reduction contributes to a significant reduction in latency. By leveraging a smaller polynomial degree N, the overall latency experiences substantial improvement.

### A.6 More Training Details and Insight

In this section, we present the training curves for the STGCN-3-256 model, which employs 6 to 1 effective second-order polynomial (non-linear) layers. Figures 7(a) through Figure 7(f) depict the training curve progression. During the training process, we utilized mixed-precision training for the polynomial model, which led to occasional instability in some iterations, as evidenced by spikes in the loss values. Nevertheless, the training process demonstrated rapid recovery following such loss spikes.

As demonstrated in training curve, a smaller number of second-order polynomial (non-linear) layers contribute to a more stable training process and facilitate smoother convergence. This finding

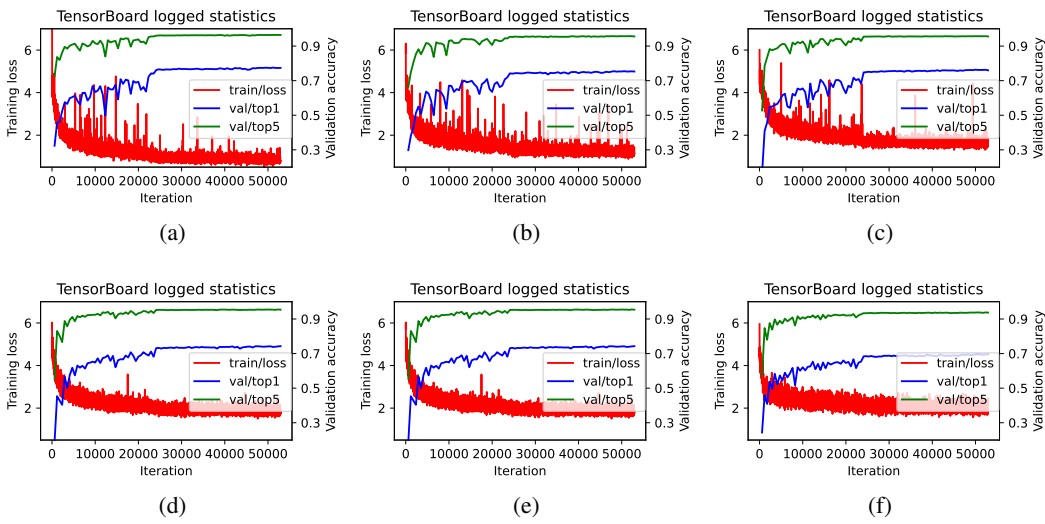

Figure 7: Polynomial replacement training curves for (a) 6-STGCN-3-128 (b) 5-STGCN-3-128 (c) 4-STGCN-3-128 (d) 3-STGCN-3-128 (e) 2-STGCN-3-128 (f) 1-STGCN-3-128

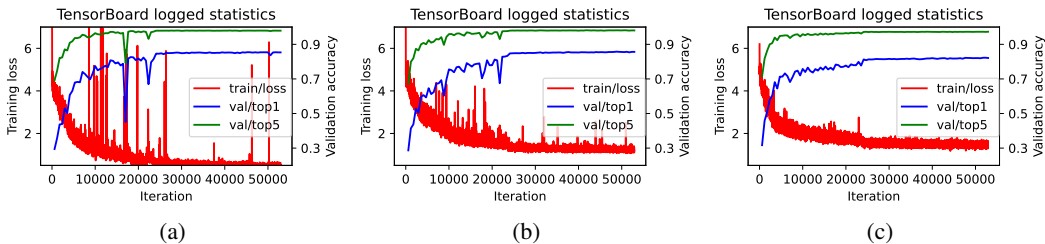

Figure 8: Polynomial replacement training curves for (a) 12-STGCN-6-256 (b) 4-STGCN-6-256 (c) 2-STGCN-6-256

explains the enhanced performance of the STGCN-6-256 model, which features a reduced number of non-linear layers, as compared to the full-polynomial model baseline.

To substantiate our hypothesis, we plot the polynomial replacement training curve for the STGCN-6-256 model in Figure 8. The training curves for 12 effective non-linear layers (12-STGCN-6-256), four effective non-linear layers (4-STGCN-6-256), and two effective non-linear layers (2-STGCN-6-256) are presented. As the number of non-linear layers increases, the model achieves greater expressivity; however, the polynomial replacement process becomes increasingly unstable. Consequently, for the STGCN-6-256 model with only 4 non-linear layers, a more stable replacement process facilitates better convergence, ultimately leading to improved accuracy performance.

