# OpenReview forum: "LinGCN: Structural Linearized Graph Convolutional Network for Homomorphically Encrypted Inference"
_NeurIPS.cc/2023/Conference — NeurIPS 2023 poster_

### Official Review · Reviewer_Y6MP · 2023-07-04

**Soundness:** 3 good
**Presentation:** 3 good
**Contribution:** 3 good
**Rating:** 6
**Confidence:** 2

**Summary:**

The paper propose a novel framework called LinGCN, which reduces the multiplication depth and optimize the performance of Homomorphic Encryption based GCN inference. According to the evaluation results, LinGCN shows promising results in both latency speedup and inference accuracy over existing approaches.

**Strengths:**

* The paper is well-written, with illustrative figures on the framework and evaluation results.
* I'm not an expert in privacy-preserving computation, but the evaluation results seem promising to me. The proposed framework not only improves the latency speedup but also the inference accuracy. Behind the stage is their proposed differentiable structural linearization, which reduces the multiplication depth in the HE process.


**Weaknesses:**

* The authors should better provide analytical results on the reduced computational cost and the utility loss. I mean although the intuition and the methodology design is convincing, some analytical statements may further support the empirical results.
* The authors should also discuss how the security of the HE-based GCN inference process is influenced with the modified designs proposed in the paper.


**Questions:**

See the weakness part above.

**Limitations:**

See the weakness part above.

---

> ### Author Rebuttal · Authors · 2023-08-09
>
> We are very grateful for your valuable insight.
>
> ### Response to weakness 1:
>
> With a reduced multiplication depth,computational cost and latency will also reduce. Analytically, the Q parameter is reduced with shallower depth, the coefficient of ciphertext polynomial also has a lower bit size, which reduces the data loading time and operators’ computational latency. Further analytical details on computation and latency reduction can be found in **Table 2** in the **appendix**.
>
> ### Response to weakness 2:
>
> The proposed LinGCN framework **does not impact** the security level of CKKS based HE framework. For all experiments with the multiplication depth reduction setting, we guarantee a fixed 128 bits margin for security level. Further details can be found in Table 1 in appendix. Here we provide a brief explanation:
>
> Initially, we have to guarantee a security level of at least 128 bits, which is determined by the encryption parameter N and Q. With one level saved, the budget of Q will also decrease.
>
> For example, with N=16384, the maximum Q=438 could guarantee a 128 bits security level. For a non-optimized 3-ST-GCN model, it required Q=509 bits to complete the HE inference (N=32768).  With one activation pruned, we save Q by 33 bits. Thus, after we pruned 3 activation layers of this model, Q = 509-33*3=410. Then, we can reduce the N from 32768 to 16384 and still guarantee a 128 bits security level.

---

> > ### Comment · Reviewer_Y6MP · 2023-08-20
> >
> > Thank you for the rebuttal. I have read the comments and will keep my score.

---

### Official Review · Reviewer_KG27 · 2023-07-05

**Soundness:** 2 fair
**Presentation:** 3 good
**Contribution:** 2 fair
**Rating:** 5
**Confidence:** 5

**Summary:**

To improve the efficiency of HE-based PPML for GCN, in this paper, the authors propose LinGCN, an end-to-end framework for non-linear reduction and polynomial replacement. LinGCN features 3 key elements, including 1) a differentiable structural linearization algorithm, 2) a compact node-wise polynomial replacement policy, and 3) finer-grained operator fusion for node-wise activation functions. The authors demonstrate good improvement over the baseline CryptoGCN.

**Strengths:**

1. The paper is well-written and the motivation is well-explained.
2. The authors demonstrate both higher accuracy and lower latency compared to the baseline CrypoGCN.
3. The ablation experiments is comprehensive.

**Weaknesses:**

1. Only one dataset is shown in the paper. This makes it doubtful for how well the proposed techniques generalize to other datasets, e.g., Cora?

2. Similar techniques have all been proposed by previous method. For example, ReLU linearization and polynomial replacement are widely studied for CNN. The proposed fusion technique is also widely used in plaintext CNN inference. Although the synchronized linearization is unique for GCN, it still makes the proposed method incremental.

3. It is unclear why LinGCN demonstrates better accuracy compared to CryptoGCN. Although KD is helpful, more analysis needs to be provided.

**Questions:**

1. What can LinGCN achieve better accuracy compared to CryptoGCN? What if similar distillation is leveraged for CryptoGCN as well?

2. How are the proposed linearization method different compared to those used for CNNs, like Selective Network Linearization (SNL), RRNet, etc.

3. How does the proposed method generalize to other datasets or tasks?

---

> ### Author Rebuttal · Authors · 2023-08-09
>
> We sincerely appreciate your insightful review.
>
> ### Response to question 1 (Combined response with weakness 3):
>
> Please refer to **Global Response (ii)** for further detail.
>
> ### Response to question 2 (Combined response with weakness 2):
>
> Thanks for the valuable insight. To the best of our knowledge, our LinGCN framework is the first attempt to apply fine-grained **structural linearization** in CKKS based HE settings for GCN to significantly reduce the multiplication depth. Although LinGCN shares some similarity between existing CNN works such as SNL and RRNet, the fundamental problem we are trying to solve is different. LinGCN focuses on leveraging **structural linearization** methods to **reduce the multiplication depth** in CKKS based HE setting, and it can benefit all HE based operators (refer to Table 2 in appendix). SNL and RRNet try to partially replace ReLU with linear or polynomial function to reduce the nonlinear operator latency of the private inference under MPC setting. And those methods in CNN works cannot benefit CKKS based HE setting, as discussed in the manuscript from line 157 to line 175. Our LinGCN is the first to significantly advance the benchmark for fine-grained multiplication depth reduction, representing a new baseline to accelerate the HE-based GCN private inference-an emerging field that has been far less explored than that in CNN. We believe our new results and insights would help the community further move forward in this important field.
>
> ### Response to question 3 (Combined response with weakness 1):
>
> The proposed LinGCN is evaluated on NTU-XVIEW skeleton joint dataset, one of the largest graph datasets which has been evaluated in CKKS based HE scheme so far. We also evaluate the performance using three model variants:STGCN-3-128, STGCN-3-256, and STGCN-6-256, where the first number represents the layer count, and the second number represents the last STGCN layer's channel count.
>
> Per reviewer request, we provide an extended evaluation of LinGCN framework on GCN model for **Flickr dataset**. The Flickr dataset consists of 89,250 nodes, 989,006 edges, and 500 feature dimensions. This dataset's task involves categorizing images based on descriptions and common online properties. Please refer to **Global Response (i)** for further details.

---

> ### Author Response · Authors · 2023-08-17
>
> Dear Reviewer,
>
> May we kindly inquire if the responses have sufficiently addressed your concerns, or if further explanations or clarifications are needed? Your time and efforts in evaluating our work are greatly appreciated.
>
> Best

---

> > ### Comment · Reviewer_KG27 · 2023-08-19
> >
> > Thank the author for the clarification.
> >
> > My concern about the novelty of linearization is still not fully addressed. While methods like SNL or RRNet cannot help reduce the multiplication depth, the main reason is on their ReLU pruning pattern, not their methods. There are other papers, e.g., "Making Models Shallow Again: Jointly Learning to Reduce Non-Linearity and Depth for Latency-Efficient Private Inference", which simultaneously reduce the ReLU counts and the depth of the network. The advantages of the proposed method compare with these works are not very clear.
> >
> > Hence, I tend to keep my original score for the paper.

---

> > > ### Author Response · Authors · 2023-08-20
> > >
> > > We very much appreciated your time and further insight and would like to provide our explanation as follows:
> > >
> > > First, thanks a lot for pointing us to the new reference [1]. However, we notice that [1] is available in arxiv on Apr 26th, and it was published in CVPR 2023 (Jun 18, 2023 – Jun 22, 2023), while this NeurIPS work was submitted on May 11th. Due to the short time, we are unable to (though we would like to) include the comparison with [1] during the paper submission and the rebuttal time right now.  Also, according to **the Neurips 2023’s policy on comparisons to recent work**-*”Papers appearing less than **two months** before the submission deadline  are generally considered concurrent to NeurIPS submissions. Authors are not expected to compare to work that appeared only a month or two before the deadline”*, [1] is considered as a concurrent work.
> > >
> > > Second, we would like to further emphasize the technique difference between ours and [1] and why our method outperforms these existing works like [1]: Work [1] removes the whole ReLU layer based on sensitivity to reduce the model depth, and focuses on MPC based PI acceleration. However, the sensitivity is obtained from the original full model, and may lead to inconsistency when nonlinear is aggressively pruned.  In principle, the method used by [1] is similar to that of the CryptoGCN [2], of which the key is the nonlinear sensitivity checking and pruning. However, as our comparison results in Figure 1 (page 2) confirm, it is less effective than our LinGCN framework. This is because our LinGCN uses gradient based structural linearization (Section 3.2, page 4-5) with L0 regularization during the training phase, which is different from and also superior to the sensitivity checking based method.
> > >
> > >
> > > We sincerely hope those explanations address your concern. We could include the quantitative comparison and discussion in the updated version based on the reviewer’s further advice.
> > >
> > > [1] Making Models Shallow Again: Jointly Learning to Reduce Non-Linearity and Depth for Latency-Efficient Private Inference
> > >
> > > [2] CryptoGCN: Fast and Scalable Homomorphically Encrypted Graph Convolutional Network Inference

---

### Official Review · Reviewer_6Lak · 2023-07-07

**Soundness:** 3 good
**Presentation:** 3 good
**Contribution:** 3 good
**Rating:** 5
**Confidence:** 3

**Summary:**

LinGCN optimizes HE-based GCN inference by reducing multiplication levels through a differentiable structural linearization algorithm and a compact node-wise polynomial replacement policy, both guided by a two-level distillation from an all-ReLU teacher model. LinGCN also improves HE solutions for GCN private inference, enabling finer operator fusion for node-wise activation functions.

**Strengths:**

1 Important and well-motivated problem

2 Experiments show a clear improvement in latency and accuracy compared to prior work.

**Weaknesses:**

While the concepts of structural linearization and the replacement of ReLU with a polynomial function are not novel in themselves, their application to Graph Convolutional Networks (GCN) is noteworthy. However, it's important to mention that the degree of novelty in this approach is not particularly significant, given the pre-existing knowledge in this field.



**Questions:**

The curves in Figure 1 is confusing. Why the blue curve encompasses a variety of points (red stars, orange dots, and blue dots)? Similarly, the black curve also exhibits this characteristic.



**Limitations:**

The paper does not discuss the limitations of the current work nor does it discuss any negative societal impact. Concentrating solely on non-linear operations may not be sufficient for achieving PPML-efficient architectures. This is particularly true in light of emerging lighter cryptographic topologies for non-linear operations, such as those discussed in the following work:

Huang, Zhicong, et al. "Cheetah: Lean and fast secure {two-party} deep neural network inference." 31st USENIX Security Symposium (USENIX Security 22). 2022.

---

> ### Author Rebuttal · Authors · 2023-08-09
>
> We are very grateful for your valuable comments.
>
>
> ### Response to weakness:
>
>
> Thanks for the important feedback. Our main contribution **structural linearization** for multiplication depth reduction is intrinsically different from existing linearization methods (unstructured linearization, layer-wise linearization) used in CNN works and CryptoGCN. **structural linearization** based multiplication depth reduction especially benefits the CKKS based HE setting. To the best of our knowledge, none of the SOTA CNN works (e.g., SNL, DELPHI) have adopted the **structural linearization** method. The polynomial replacement itself is only one of the contributions in our LinGCN framework. Our major contributions are summarized in **Global Response (ii)**.
>
> ### Response to question:
>
>
> Thanks for the suggestions. As indicated in the figure legend, blue curve is the pareto frontier of the LinGCN framework. Red stars means STGCN-3-128 model used in LinGCN framework. Orange dots means STGCN-3-256 model used in LinGCN framework, and blue dots means STGCN-6-256 model used in LinGCN framework. The black dots and black curve represent the model performance data points collected from CryptoGCN and its pareto frontier. We will include more descriptions in the legend and make the figure clearer in the final version.
>
> ### Response to limitations:
>
>
> Cheetah requires the client's assistance while conducting computation because it uses a MPC + HE  based framework. While our work’s setting does not require the client’s assistance because we use CKKS based HE scheme for private inference. The ciphertext multiplication method used in [1] is BFV, in which it requires the client to do the decryption for the extracted LWE ciphertext, re-encryption for intermediate results and send the new ciphertexts to the server, then the server can perform the next layer’s computation. Client computation and communication are the bottleneck in their proposed method [1]. The major latency bottleneck  in our CKKS based HE system is the **multiplication depth** which is associated with the linear and nonlinearity operation, and is not the nonlinearity itself. Hence, structural linearization helps with reducing the ring size while maintaining the same security level, and will make all operator latency to be smaller. The detailed operator latency breakdown with **multiplication depth reduction** can be found in the Appendix Table 2. The table shows the rotation is the major performance bottleneck, not the nonlinearity operator itself.
>
>
> [1] Huang et al., "Cheetah: Lean and fast secure two-party deep neural network inference," USENIX Security 2022.

---

### Official Review · Reviewer_iVPr · 2023-07-08

**Soundness:** 3 good
**Presentation:** 4 excellent
**Contribution:** 3 good
**Rating:** 7
**Confidence:** 4

**Summary:**

This study presents an approach for enhancing the efficiency of private inference ( using homomorphic encryption (HE)) in Spatiotemporal graph convolutional networks through fine-grained and structured pruning/dropping of non-linearity. The proposed method consists of two main steps. Firstly, ReLUs are eliminated by substituting them with identity/linear operations, reducing the multiplicative depth required for HE operations. Subsequently, the remaining non-linearities are replaced with HE-friendly quadratic activations.

**Strengths:**

1. The idea of optimizing ReLUs with a fine-grained and structured approach is promising. Typically, fine-grained optimizations tend to lack structure, which can reduce the overall benefits of optimization.

2. The paper is well-structured and easy to understand. The proposed method is explained clearly and organized, making it accessible to readers. The authors provide detailed explanations and build a strong foundation for their approach, ensuring it can be easily grasped.

**Weaknesses:**



**Limited novelty (compared to CryptoGCN)**:
Compared to the CryptoGCN paper (NeurIPS'22), this paper demonstrates limited novelty. Both papers propose techniques to reduce the multiplicative depth in STGCN and replace ReLUs with quadratic activations. However, while CryptoGCN employs fine-tuning to recover the accuracy drop, this paper utilizes logit and feature-based distillation to address the accuracy drop resulting from the reduction in non-linearity.

For a fair comparison, particularly regarding the improvement in accuracy,  an ablation study should be included in the paper to demonstrate the benefits of logit-based knowledge distillation, feature-based knowledge distillation, and the combined usage of both techniques compared to CryptoGCN.


**CKKS implementation is used rather than rotation-free HE**
In this paper, the authors used the CKKS implementation of Homomorphic Encryption (HE), which incurs a significant total latency due to the rotation operations. The provided Table 2 in the Appendix demonstrates that **approximately 90%** of the total HE latency is consumed by rotation operation in the 12-STGCN-6-256 model. This highlights the inefficiency of the CKKS implementation for private inference, especially when considering the availability of rotation-free HE techniques [1] and their implementation in Microsoft-SEAL.

**Minor corrections**
As stated in lines #157 to #161, the optimizations implemented in SNL, DELPHI, and SAFENet are sub-optimal for achieving high efficiency in HE. It is important to note that these optimizations primarily aim to minimize the cost of non-linearity in Multi-Party Computation, under the assumption that intensive HE operations can be carried out offline.




1. Huang et al., "Cheetah: Lean and fast secure two-party deep neural network inference," USENIX Security 2022.



**Questions:**

1. The sensitivity analysis (Figure 5) shows that ReLUs in the penultimate layers (layer 4) are highly effective, which aligns with similar findings in CNN studies like DeepReDuce (ICML'21) and SENet (ICLR'23). What does it imply about feature learning within CNNs vs GCNs?


2. Is the better accuracy of LINGCN over CryptoGCN only due to CryptoGCN's use of two-level Knowledge Distillation? Both these methods use second-degree polynomial activation (instead ReLUs).

3. Does the accuracy mentioned in the paper refer to floating-point accuracy (in plaintext) or fixed-point accuracy (after converting weights/biases and activation to fixed-point for operations in the ciphertext domain)?



**Note** I am open to increasing the score if most concerns are addressed in the rebuttal.

**Limitations:**

Limitations are not discussed in the paper.

---

> ### Author Rebuttal · Authors · 2023-08-09
>
> We are very grateful for your constructive comments.
>
>
> ### Response to limitation 1, novelty:
>
>
> CryptoGCN's approach to employing layer-wise nonlinear pruning and polynomial replacement results in a substantial degradation in accuracy and an insufficient reduction in multiplication depth. Our proposed LinGCN framework is different from CryptoGCN. LinGCN employs: (i) **Differentiable structural linearization**. (ii) Node-wise polynomial activation functions augmented by distillation. (iii) Exploration of nonlinear reduction via polynomial sequences. (iv) Homomorphic Encryption (HE) for enabling finer-grained operator fusion.The LinGCN is designed to significantly reduce multiplication depth and yield much better performance compared to CryptoGCN.
>
>
> The effects of these strategies have been empirically demonstrated in the ablation study, as illustrated in Figure 6 (Page 9), which encompasses the (a) replacement sequence, (b) node-wise or layer-wise nonlinear reduction, (c) logit-based distillation, and (d) feature-based distillation.
>
>
>
> ### Response to limitation 2, encryption scheme:
>
>
> Our research has a different threat model setting in comparison to the setting described in [1]. Specifically, the rotation-free Homomorphic Encryption (HE) described in [1] requires client assistance for private inference, assuming that clients have significant computational capabilities and can actively participate in the encrypted computation process. This method, which leverages the BFV scheme for ciphertext multiplication, requires the client to decrypt the extracted LWE ciphertext, re-encrypt intermediate results, and transmit new ciphertexts to the server for subsequent layer computation. Overall, the framework in [1] introduces computational and network communication bottlenecks from the client side.
>
>
> In contrast, our methodology is focused on an HE without-client-aid setting, in which the server solely requires the client to encrypt and transmit input data once. Then, the server autonomously conducts the necessary computations.
>
>
> Our HE without-client-aid setting and the MPC+HE setting in [1] represent two orthogonal strategies for implementing Privacy-Preserving Machine Learning (PPML). Each approach caters to distinct private inference scenarios. The primary objective of our LinGCN framework is to alleviate the server's computational burden within the context of an HE without-client-aid setting.
>
>
> [1] Huang et al., "Cheetah: Lean and fast secure two-party deep neural network inference," USENIX Security 2022.
>
>
> ### Response to limitation 3, minor corrections:
>
>
> Thanks for the suggestion, existing works such as SNL, DELPHI, and SAFENet focus on latency reduction of online inference with MPC-only framework, and those optimizations can not directly benefit the CKKS based HE for offline inference framework. We will clarify it further in the final version.
>
>
> ### Response to question 1:
>
> The results of our experiment indicate that the last layer's nonlinearity is more prone to be linearized, whereas the 4th layer serves as the most vital nonlinear layer. This observation is consistent with works in CNN, as evidenced in studies such as SNL, DeepReDuce, and SENet. Within these models, the importance trend of nonlinearity layers appears to follow a pattern that fluctuates from low to high and then reverts to low across the network layers.
>
> We further have conducted evaluation on a new graph dataset (i.e., Flickr). Pelase referr to the **Global Response (i)** for more details. We will incorporate those discussions in the final version.
>
> ### Response to question 2:
>
> Please refer to the **Response to limitation 1** and **Global Response (ii)**
>
> ### Response to question 3:
>
> The accuracy mentioned in our paper indicates the floating-point accuracy (in plaintext). We use the 2^33 scaling factor to transform the input into fixed points for encoding & encryption settings, which is the same as SOTA, e.g., CryptoGCN. As mentioned in CryptoGCN, the large scaling factor and  bit width margin makes sure the ciphertext inference has the same accuracy as plaintext. We will include further discussion about plaintext/ciphertext encoding and accuracy in the final version.

---

> > ### Comment · Reviewer_iVPr · 2023-08-11
> > **Rebuttal's response**
> >
> >
> > Thank you for the rebuttal, and I acknowledge the Author's effort in producing results on the new dataset. I'm convinced with the ablation study presented in Figure 6c and Figure 6d, and it is evident that the accuracy benefits (over CryptoGCN) of the proposed method mainly stem from the (relatively) loss-less reduction to multiplicative depth.
> >
> > I have *only one question* about the study presented in Figure 5 (an extension to the previously asked Q1): What is Authro's intuition behind Layer 4's criticality being the highest? Figure 5 is a **good observation**; however, the Authors should have presented the **insights** for the same. In general, the semantic information presented in a layer of a network increases from the initial layer to the deeper layer, and the redundancy in features is higher in the deeper layer.

---

> > > ### Author Response · Authors · 2023-08-13
> > > **Insights and Intuition Behind Figure 5: Importance of Nonlinear Layers**
> > >
> > > We appreciate your thoughtful comments and your recognition of our efforts.
> > >
> > >
> > > The model in Figure 5 is STGCN (STGCN-3-256), with 3 STGCN layers and 6 nonlinear layers. The nonlinear layers 1-2 reside in the first STGCN layer. Nonlinear layers 3-4 reside in the second STGCN layer. Nonlinear layers 5-6 reside in the last STGCN layers.
> > >
> > > The STGCN layer employs the GCNConv operator to capture the relationships and information among the node neighbor features [1]. Unlike CNNs which may benefit from deeper layers in most cases, GCNs exhibit a performance limit of how many layers we can stack [1]. Too many stacked GCN layers may lead to node representations becoming too similar. The phenomena is called **over-smoothing** [1]. For GCNs with too deep layers, as each layer aggregates information from neighboring nodes, distant nodes start to influence each other, potentially resulting in loss of model expressiveness and diminished performance [1]. The over-smoothing problem might be exacerbated if nonlinear layers are extremely linearized. Intuitively,  it is preferred to preserve nonlinearity for every several linearized GCN layers to prevent the over-smoothing effect from getting worse.
> > >
> > > In Figure 5, the 4th nonlinear layer is situated at the second (middle) STGCN layer of STGCN-3-256, which may be crucial for mitigating the over-smoothing problem associated with prior deep linearized STGCN layers. During the automatic structural linearization process, the gradient propagation is orchestrated to strike a **balance between two goals**: (i) preserving the feature structural information at the deeper layers, similar to techniques used in CNN based model (SNL[2], DeepReduce[3], and SENet[4]), and (ii) mitigating the **over-smoothing** effect resulting from the deep linearized GCN layers. Consequently, the middle layer's (nonlinear layer 4 in Figure 5) nonlinearity emerges as the most vital component within the STGCN architecture. This phenomenon is also observable in the STGCN-6-256 model nonlinear reduction result, where we observe that  the 5th and 6th nonlinear layers (reside within the 3rd STGCN layer) are most important for the network's nonlinearity.
> > >
> > > We will incorporate this discussion and corresponding references into the final version.
> > >
> > > [1] Simple and Deep Graph Convolutional Networks, ICML, 2020
> > >
> > > [2] Selective Network Linearization for Efficient Private Inference, ICML, 2022
> > >
> > > [3] DeepReduce: ReLU Reduction for Fast Private Inference, ICML, 2021
> > >
> > > [4] Learning to Linearize Deep Neural Networks for Secure and Efficient Private Inference, ICLR, 2023

---

> > > > ### Comment · Reviewer_iVPr · 2023-08-14
> > > > **Reply to Author's comment**
> > > >
> > > > Thank you for providing the intuition behind the importance of non-linear layers in STGCN. I would like to recommend that the Authors incorporate the following aspects in their subsequent paper iteration:
> > > >
> > > > 1) Criticality analysis of non-linear layers on STGCN-6-256.
> > > > 2) Impact of two-layer KD on the importance of non-linear layers in  STGCN. In particular, does the importance of non-linear layers remain the same or alter when employing two levels of KD?
> > > > 3) A discussion on how the relevance of non-linear layers differs in GCN compared to CNN. Considering that both CNN and GCN's initial layers identify local patterns, why do non-linearities in early STGCN layers appear more crucial than in CNN layers (as shown in DeepReDuce, SNL, and SENet, Stage 1 non-linearity is least critical)? Does this suggest that local patterns captured by the early layers in GCN contribute more to the network's discriminative power compared to those in CNN?
> > > >
> > > >
> > > > After considering the rebuttal and the subsequent discussion, I have raised the score to Accept.

---

> > > > > ### Author Response · Authors · 2023-08-14
> > > > >
> > > > > We are grateful for your insightful remarks. We'll include prior discussions and address those further comments in the final version of the paper. Thank you for helping us improve the quality of the paper!

---

### Author Rebuttal · Authors · 2023-08-09

## Global Response:


We truly appreciate your valuable and constructive comments. We have made a substantial effort to clarify your doubts and enrich our experiments in the rebuttal phase. Below are the responses to two common doubts:


### (i). New dataset evaluation:


Without loss of generality，we extended our evaluation on flickr dataset, which is a representative node classification dataset widely used in GNN tasks.It consists of 89,250 nodes, 989,006 edges, and 500 feature dimensions. This dataset's task involves categorizing images based on descriptions and common online properties.


For the security setting, we assume that node features are user-owned and the graph adjacency list is public. The Flickr dataset has a larger adjacent list but smaller feature dimension compared to the NTU-XVIEW dataset. We utilize three GCN layers with 256 hidden dimensions. Each GCN layer has similar structure as STGCN used in our paper. We conduct full-batch GCN training to obtain ReLU-based baseline model accuracies of 0.5492/0.5521 for validation/test dataset.


We obtain the accuracy/latency tradeoff detailed in the following table.


| Num. of nonlinear layers  in GCN layers | Accuracy (val/test, %) | Latency (s) |
|:---------------------------------------:|:----------------------:|:-----------:|
|                    6                    |      0.5281/0.5275     |     4290.93252s     |
|                    2                    |      0.5247/0.5266     |     2740.93779s     |
|                    1                    |      0.5269/0.5283     |     2525.79771s     |


We observe that the proposed LinGCN framework substantially diminishes the number of effective nonlinear layers and thus reduces the multiplication depth, with comparable accuracy. This leads to an expedited private inference within the GCN model (1.7 times speedup). The experiments on the  Flickr dataset substantiate the generalization capability of the LinGCN framework, reinforcing the robustness of our proposed methods.


### (ii) Contributions and ablation study:


The superior performance of the LinGCN framework over CryptoGCN is not solely attributable to KD. As Figure 6(c)(d) (page 9) in the ablation studies reveal, a two-level KD process contributes to an approximate 1% improvement in accuracy. The LinGCN framework achieves both enhanced accuracy and reduced multiplication depth through a multifaceted approach that includes: (i) **Differentiable structural linearization**. (ii) Node-wise polynomial activation functions, coupled with distillation. (iii) Nonlinear reduction through polynomial sequence exploration. (iv) Extended Homomorphic Encryption (HE) solutions to enable finer-grained operator fusion.


Collectively, these contributions render the LinGCN's performance markedly superior to CryptoGCN. Figure 6 in the Ablation study further elucidates the effects of these factors on the results, showcasing the (a) replacement sequence, (b) node-wise or layer-wise nonlinear reduction, (c) logit-based distillation, and (d) feature-based distillation.

---

### Decision · Program_Chairs · 2023-09-21

**Decision:**

Accept (poster)

**Comment:**

The paper received 4 reviews, and the authors did a solid job in addressing all the reviewer comments and suggestions. This includes running new experiments to analyze performance and run-time. Overall, the general view of the reviewers was quite positive and leaning to accept. They generally commended the authors on proposing a well-founded method and providing a fresh perspective on the subfield of GCNs. The rebuttal seems to have addressed all reviewer concerns.